# A Multi-Omics and Human Biomonitoring Approach to Assessing the Effectiveness of Fortified Balanced Energy–Protein Supplementation on Maternal and Newborn Health in Burkina Faso: A Study Protocol

**DOI:** 10.3390/nu15184056

**Published:** 2023-09-19

**Authors:** Yuri Bastos-Moreira, Lionel Ouédraogo, Marthe De Boevre, Alemayehu Argaw, Brenda de Kok, Giles T. Hanley-Cook, Lishi Deng, Moctar Ouédraogo, Anderson Compaoré, Kokeb Tesfamariam, Rasmané Ganaba, Lieven Huybregts, Laeticia Celine Toe, Carl Lachat, Patrick Kolsteren, Sarah De Saeger, Trenton Dailey-Chwalibóg

**Affiliations:** 1Center of Excellence in Mycotoxicology and Public Health, MYTOXSOUTH^®^ Coordination Unit, Faculty of Pharmaceutical Sciences, Ghent University, 9000 Ghent, Belgium; marthe.deboevre@ugent.be (M.D.B.); sarah.desaeger@ugent.be (S.D.S.); 2Department of Food Technology, Safety and Health, Faculty of Bioscience Engineering, Ghent University, 9000 Ghent, Belgium; lionel.olivier.ouedraogo@gmail.com (L.O.); alemayehuargaw.alemayehu@ugent.be (A.A.); brenda.dekok@gmail.com (B.d.K.); giles.hanleycook@ugent.be (G.T.H.-C.); lishi.deng@ugent.be (L.D.); kokebtesfamariam.hadush@ugent.be (K.T.); l.huybregts@cgiar.org (L.H.); laeticiaceline.toe@ugent.be (L.C.T.); carl.lachat@ugent.be (C.L.); patrick.kolsteren@ugent.be (P.K.); 3Centre Muraz, Bobo-Dioulasso 01 BP 390, Burkina Faso; 4Agence de Formation de Recherche et d’Expertise en Santé pour l’Afrique (AFRICSanté), Bobo-Dioulasso 01 BP 298, Burkina Faso; obmoctar@gmail.com (M.O.); discompa4523@gmail.com (A.C.); rganaba@hotmail.com (R.G.); 5Nutrition, Diets, and Health Unit, Department of Food and Nutrition Policy, International Food Policy Research Institute (IFPRI), Washington, DC 20005, USA; 6Unité Nutrition et Maladies Métaboliques, Institut de Recherche en Sciences de la Santé (IRSS), Bobo-Dioulasso 01 BP 545, Burkina Faso; 7Department of Biotechnology and Food Technology, Faculty of Science, University of Johannesburg, Doornfontein Campus, Gauteng 2028, South Africa

**Keywords:** balanced energy–protein supplementation, metabolomics, proteomics, metagenomics, exposome, MISAME-III

## Abstract

Fortified balanced energy–protein (BEP) supplementation is a promising intervention for improving maternal health, birth outcomes and infant growth in low- and middle-income countries. This nested biospecimen sub-study aimed to evaluate the physiological effect of multi-micronutrient-fortified BEP supplementation on pregnant and lactating women and their infants. Pregnant women (15–40 years) received either fortified BEP and iron–folic acid (IFA) (intervention) or IFA only (control) throughout pregnancy. The same women were concurrently randomized to receive either a fortified BEP supplement during the first 6 months postpartum in combination with IFA for the first 6 weeks (i.e., intervention) or the postnatal standard of care, which comprised IFA alone for 6 weeks postpartum (i.e., control). Biological specimens were collected at different timepoints. Multi-omics profiles will be characterized to assess the mediating effect of BEP supplementation on the different trial arms and its effect on maternal health, as well as birth and infant growth outcomes. The mediating effect of the exposome in the relationship between BEP supplementation and maternal health, birth outcomes and infant growth were characterized via biomonitoring markers of air pollution, mycotoxins and environmental contaminants. The results will provide holistic insight into the granular physiological effects of prenatal and postnatal BEP supplementation.

## 1. Background

In many low- and middle-income countries (LMICs), pregnant women struggle to meet the nutritional requirements for sustaining healthy fetal development [1] (UNICEF-WHO, 2019); left unmet, these crucial needs can result in miscarriage [2] (Benammar et al., 2012) and stillbirth [3] (McClure et al., 2009). Born alive, these infants are often of low birth weight (LBW), small for gestational age (SGA) and/or preterm, and they experience severe health and developmental disadvantages, resulting in undue costs to society [4] (Almond et al., 2004).

In 2016, to reduce the risk of stillbirths and SGA births, the WHO published antenatal care guidelines recommending balanced energy–protein (BEP) supplementation during pregnancy for all pregnant women in the context of high prevalenceof population-level undernutrition [5] (WHO, 2016). This recommendation, however, was based on evidence of “moderate certainty” and suggests that BEP supplementation “probably” reduces the rates of stillbirth and SGA birth [5,6] (Ota et al., 2012; WHO, 2016). In an LMIC, any costly antenatal intervention (like BEP supplementation) supported by evidence of “moderate certainty” is unlikely to be prioritized, at least until more robust research and evidence-based data can demonstrate a return on investment. As a result, the Bill & Melinda Gates Foundation (BMGF) built and funded a large-scale multi-centric consortium of seven intervention studies aimed at evaluating the effectiveness of BEP supplementation during pregnancy and lactation [7] (Gernand et al., 2023).

The protocol of the MISAME-III main trial was published previously [8] (Vanslambrouck et al., 2021). The study was a community-based, non-blinded, individually randomized 2 × 2 factorial randomized controlled trial (RCT) involving directly observed daily supplement intake (Figure 1). The primary outcomes were the prevalence of SGA at birth (<10th percentile of the newborns size standards of the International Fetal and Newborn Growth Consortium for the 21st Century (INTERGROWTH-21st) [9] (Villar et al., 2013)) and the length-for-age z-score (LAZ) at 6 months of age (calculated using the WHO’s 2006 growth reference at 6 months of age [10] (de Onis and Branca, 2016)). These results have been published [11,12,13] (de Kok et al., 2022; Argaw, de Kok, et al., 2023; Argaw, Toe, et al., 2023).

The secondary and exploratory biological outcomes of the prenatal BEP intervention were maternal and newborn body compositions and newborn relative telomere length (TL), with mitochondrial DNA content (mtDNAc) a non-declared outcome that was considered relevant for the trial at the time at which the samples were analyzed. The former has been published [12] (Argaw, Toe, et al., 2023), and the latter is under review [14] (Hanley-Cook et al., no date).

Anthropometry serves as a valuable metric, enabling the comparison of an individual child to a growth reference derived from a healthy population [15] (Perumal, Bassani and Roth, 2018). However, these are relatively crude metrics, and biomarkers are required to fully characterize the physiological changes hypothesized to result from maternal perinatal BEP supplementation [12,13] (de Kok et al., 2022; Argaw et al., 2023).

Thus, this dedicated biospecimen sub-study (BioSpé), which was nested within the larger MISAME-III trial, aimed to evaluate the physiological effects of multi-micronutrient-fortified BEP supplementation on 309 pregnant and lactating women (PLW) and their infants (Table 1).

Due to a dearth of evidence on maternal and infant physiology during maternal BEP supplementation, we adopt an unbiased, hypothesis-generating approach aimed to uncover biological pathways and discover novel biomarkers for assessing maternal health (gestational weight gain and anemia [16] (Hanley-Cook, Toe, et al., 2022)) as well as infant development and growth (Figure 2).

## 2. Methods: Study Design, Biospecimen Collection and Rationale

### 2.1. Study Setting, Participants and Enrolment Procedures

The MISAME-III trial was implemented in 6 rural health-center catchment areas in Burkina Faso. The usual diet during pregnancy is predominantly maize-based, with the addition of leafy vegetables [17] (de Kok, Argaw, et al., 2021). The transmission of malaria is continual with seasonal variations [18] (Hanley-Cook, Argaw, et al., 2022).

PLW aged between 15 and 40 years who lived in the study villages were identified through a census conducted in the research area (*n* = 10,165). Community support staff visited all eligible participants at their residences every five weeks to identify early pregnancy by screening for self-reported amenorrhea. Women who were suspected of being pregnant were guided to the health center for a urine pregnancy test, and pregnancies were confirmed via ultrasounds. The participants excluded from the study included those who planned to leave the study area during pregnancy or deliver outside the study area and individuals who had a peanut allergy because BEP is an energy-dense peanut paste.

After written informed consent was provided, the participants were randomly assigned to the prenatal intervention arms receiving either the fortified BEP supplements and iron–folic acid (IFA) tablets (i.e., intervention) or the IFA tablets alone (i.e., control), which is the standard of care during pregnancy. The same participants were concurrently randomized to one of two of the postnatal intervention arms, either receiving fortified BEP supplementation during the first 6 months postpartum in combination with IFA for the first 6 weeks (i.e., intervention), or IFA alone for 6 weeks postpartum (i.e., control). Therefore, the participants were randomized into one of the 4 study groups: (1) both pre- and postnatal BEP and IFA supplementation (BEP/BEP); (2) prenatal BEP supplementation and postnatal IFA supplementation only (BEP/IFA); (3) prenatal IFA supplementation and postnatal BEP and IFA supplementation (IFA/BEP); or (4) both pre- and postnatal IFA supplementation (IFA/IFA).

Trained village-based project workers visited the pregnant participants to observe their intake of the BEP supplement and IFA tablets. When the participants were absent from home, the BEP and IFA were provided in advance (thus counting as non-observed intakes). The participants were encouraged to attend scheduled antenatal care (ANC) visits every seven weeks per national policy [19] (Ministère de la Santé—Burkina Faso, 2010). 

### 2.2. Study Supplements

A formative study was conducted to establish a preferred and suitable supplement according to the Burkinabè population [20] (Jones et al., 2021). The chosen supplement was an energy-dense peanut paste fortified with multiple micronutrients. Table 2 [21] (Bill & Melinda Gates Foundation, 2017) provides the nutritional composition of the BEP.

The PLW in the intervention group received a daily BEP supplement and an IFA tablet (Sidhaant Life Sciences, Delhi, India) containing 65 mg of iron (in the form of FeH_2_O_5_S) and 400 μg of folic acid (in the form of C_19_H_19_N_7_O_6_; the tolerable upper intake level from fortified food or supplements, not including folate from food, is 1000 μg/d [22] (Allen, Carriquiry and Murphy, 2020), whereas those in the control group received a daily IFA tablet only, in accordance with the standard of care in Burkina Faso. Following Burkinabè guidelines, during ANC visits, all participants received malaria prophylaxis (three oral doses of sulfoxide–pyrimethamine).

### 2.3. Biospecimen Collection

#### 2.3.1. Whole Blood (Mother–Infant Dyads)

A total of 60 µL (2 × 10 µL and 2 × 20 µL) of whole blood was collected via capillary sampling, using a volumetric absorptive microsampling (VAMS) device. The VAMS technology wicks a small, fixed volume of biofluid, which is beneficial for newborns and anemic participants, and poses fewer challenges in the handling, storage and transportation of samples [23] (Vidal et al., 2021). Both 10 µL Mitra Clamshell (2-sampler) and 20 µL Mitra Clamshell (2-Sampler) devices (item numbers: 10,109 and 20,109), namely Mitra^TM^, were obtained from Neoteryx (Torrance, CA, USA).

Samples were collected via the VAMS device at the following timepoints: trimester 2 (19–24 weeks of gestation), trimester 3 (30–34 weeks of gestation) and 5–6 months (147–175 days) postpartum in the PLW participants. In infants, 60 µL of whole blood was collected via the VAMS device at birth and 1–2 months (28–56 days of life), 3–4 months (84–112 days of life) and 5–6 months (140–168 days) postnatally. 

The VAMS technology wicks a small, fixed volume of biofluid, which is beneficial for newborns and anemic participants, and poses fewer challenges in the handling, storage and transportation of samples [23] (Vidal et al., 2021). Untargeted metabolomics and mycotoxin analyses were performed on 10 µL and 20 µL VAMS devices, respectively. The VAMS devices (MitraTM) were obtained from Neoteryx (Torrance, CA, USA). To preserve the integrity of the metabolites for the metabolomics analyses, the PLW were asked to not eat breakfast on the morning of the sample collection via the VAMs device. The samples collected via the VAMS device were stored in Mitra Autoracks (96-Sampler, item number: 108) inside a storage solution, using a 96-well plate packed with desiccant bags (item number: AC-SS02); both items were obtained from Neoteryx (Torrance, CA, USA) for long-term storage at −80 °C.

#### 2.3.2. Plasma (Mothers)

A total of 500 µL of whole blood was collected via capillary sampling, using plastic BD microtainer whole-blood tubes which were spray-coated with K2 potassium salt of ethylene diamine tetra acetic acid (EDTA) (BD, Franklin Lakes, NJ, USA). Following centrifugation using a microcentrifuge (VWR International, Leuven, Belgium), 100 µL of plasma was aliquoted into sterile cryotubes (Biosigma, Cona, VE, Italy) via single-channel pipettes (Thermo Fisher Scientific, Merelbeke, Belgium) before it was flash-frozen in 12 L liquid nitrogen storage vessels (Cryopal, Air Liquide, Paris, France) and transferred to a −80 °C freezer. In the PLW, the plasma samples were collected in trimester 2 (19–24 weeks of gestation) and trimester 3 (30–34 weeks of gestation) and 1–2 (28–56 days) and 5–6 months (140–168 days) postnatally.

Plasma was collected as it encompasses a broad spectrum of proteins often utilized as biomarkers to detect the biological pathways influenced by supplementation [24,25] (Weissinger et al., 2006; Chakrabarti et al., 2020).

#### 2.3.3. Cord Blood 

Within 30 min of birth, arterial umbilical cord blood was collected in 4 mL BD Vacutainer^®^ plastic whole-blood tubes which were spray-coated with K2 potassium salt of EDTA (BD, Franklin Lakes, NJ, USA). These tubes were gently inverted at least 10 times to thoroughly mix the blood with the anticoagulant. Using micropipettes (Thermo Fisher Scientific, Merelbeke, Belgium), the blood samples were aliquoted into sterile cryotubes and flash-frozen before they were transferred to a −80 °C freezer. 

Umbilical cord blood demonstrates lower biovariability and excellent DNA yield and quality [26] (Lin et al., 2019). Consequently, the impact of the BEP and environmental contaminants on newborn relative TL and mtDNAc and the presence of black carbon (BC) particles will be analyzed using the whole arterial blood collected from the umbilical cord.

#### 2.3.4. Urine (Mothers)

First-morning-void urine samples were collected from the participants in sterile 60 mL polypropylene containers (Corning Gosselin SAS, Borre, France) and aliquoted into 5 mL sterile cryotubes (VWR International, Leuven, Belgium), using 1 mL Pasteur pipettes (Deltalab, Heusden-Zolder, Belgium), before they were flash-frozen and stored at −80 °C. The first morning voids were collected to standardize urine collection between participants. The urine samples were collected in trimester 2 (19–24 weeks of gestation) and trimester 3 (30–34 weeks of gestation).

Urine is the most appropriate biological matrix for measuring acute, nonpersistent chemical exposures with rapid half-lives [27,28] (Barr et al., 2005; Esteban and Castaño, 2009); thus, analyses of the prenatal concentrations of pesticides, insecticides and herbicides will be performed on these samples.

#### 2.3.5. Breast Milk (Mothers)

Using an electric breast pump (Medela, Baar, Switzerland), a total of 7.2 mL of breast milk was aliquoted into 4 × 2 mL sterile cryotubes. This sample was drawn from a full expression of the breast adjacent to the breast last used to feed the infant, and the samples were gently inverted to homogenize fore- and hindmilk. Milk samples were collected at the following timepoints: 14–21 days after delivery and 1–2 months (28–56 days of life) and 3–4 months (84–112 days of life) postpartum.

In addition to macronutrients and micronutrients, breast milk also provides numerous bioactive components, including antibacterial peptides, antibodies, cells and microbes [29,30] (Walker and Iyengar, 2014; Ma et al., 2020). These components have an influence on the growth of the newborn and on the development of organs and systems [31,32,33,34,35,36] (Bardanzellu, Fanos and Reali, 2017; Bardanzellu et al., 2018, 2019; Congiu et al., 2019; Bardanzellu, Peroni and Fanos, 2020; Bardanzellu, Reali, et al., 2020). The milk proteome is composed of proteins and endogenous peptides [37] (Dallas et al., 2013). Proteomic studies have reported almost 3000 proteins in human milk [38] (Van Herwijnen et al., 2016).

In this study, breast milk will be analyzed to assess the multi-omics profiles between different arms of the trial [39,40] (Kisuse et al., 2018; Nguyen et al., 2021).

#### 2.3.6. Feces (Mother–Infant Dyads)

Maternal fecal samples (8 g) were collected in a fecal pot and then aliquoted into sterile cryotubes (Biosigma, Cona, VE, Italy), flash-frozen and transferred to a −80 °C freezer. Infant feces (8 g) were collected using a 38 × 50 cm sterile protection sheet (Kimberley-Clark, Irving, TX, USA) which is used like a diaper and wrapped around the newborn, before they were transferred to a OMNIgene•GUT OM-200 collection kit and then sterile cryotubes for storage at −80 °C. The collected feces were assessed for consistency based on the visual Bristol scale. For liquid feces, thorough homogenization was performed using a plastic spoon so that the solid and liquid components were mixed well.

Fecal samples were collected from the PLW in trimester 2 (19–24 weeks) and trimester 3 (30–34 weeks) and 1–2 (28–56 days) and 5–6 months (147–175 days) postpartum, and fecal samples were collected from the infants 1–2 months (28–56 days of life), 3–4 months (84–112 days of life) and 5–6 months (140–168 days) postnatally. Since feces are non-invasive, biologically rich matrixes containing host, microbe and dietary proteins [41] (Gonzalez, Zhang and Elias, 2017), these samples will be analyzed to assess the gut microbiota profiles of the participants and markers of inflammation.

### 2.4. Rationale for the Analysis of Biospecimens with Related Bio-Measurements

We will apply integrated multi-omics approaches to comprehensively characterize the metabolome, microbiome and proteome of the whole blood, plasma, breast milk and feces during pregnancy and the period of exclusive breastfeeding in both mothers and infants. A summary of the analyses and the laboratories used are shown in Table 3.

The multi-omics approaches applied will compare postnatal maternal and infant data among and between supplementation groups to identify any differences in the compositions associated with the pre- and postnatal supplementation of BEP. In addition, these analyses will aim to identify maternal and infant features are associated with infant phenotypes of adverse birth outcomes (i.e., LBW, neonatal mortality, SGA, preterm birth, stunting, underweight, wasting and underweight), as well as those correlated with continuous metrics of birth anthropometry (i.e., birth length and weight and chest, head and mid-upper arm circumferences) and with continuous metrics of growth (i.e., infant length and weight and chest, head and mid-upper arm circumference) throughout the period of postnatal follow-up (from birth to 6 months of age). 

This section describes selected analyses and the rationale and specific analytical techniques that will be employed.

#### 2.4.1. Metabolomics

Metabolomics identifies and characterizes changes in the metabolites in a biofluid (e.g., blood). Previously, a targeted metabolomics approach was used to measure essential amino acids and other metabolites in 313 Malawian children. The results reported that sixty-two percent of Malawian children with stunting had lower serum concentrations of all essential amino acids in contrast with non-stunted children, as well as lower serum concentrations of conditionally essential amino acids, non-essential amino acids and six sphingolipids and variations in the concentrations of glycerophospholipids [53] (Semba et al., 2016). Likewise, a study by Hemp et al. (2019) [54] found that the breast milk of mothers with stunted infants, in comparison to milk from mothers with a body mass index higher than 18.5, was lower in 6 amino acids/biogenic amines [54] (Hampel et al., 2019).

In the present study, we will apply untargeted metabolomics approaches to analyze maternal and infant blood, as well as targeted and untargeted metabolomics approaches to analyze breast milk. The untargeted metabolomics analyses will be conducted via rapid liquid chromatography–mass spectrometry (rLC-MS), using a previously developed method [42] (Villar et al., 2022), and targeted metabolomics analyses will be conducted using LC-MS-MS and FIA-MS/MS [52] (Langsdorf et al., 2023).

#### 2.4.2. Metagenomics

Early life is a period during which the infant gut microbiome is established, ultimately influencing health and disease later in life [55] (Tanaka and Nakayama, 2017). An important factor affecting the development and composition of the infant gut microbiome is nutrition [56,57,58] (Marques et al., 2010; Bäckhed et al., 2015; Gritz and Bhandari, 2015). Human milk has been demonstrated to induce differences in the composition of the microbiota [58,59] (Marques et al., 2010; Gomez-Llorente et al., 2013). It is established that an inadequate maturation of the gut microbiome can lead to the development of child malnutrition, both moderate and severe [60] (Vray et al., 2018).

To understand this, inStrain will be used to profile fecal metagenomes recovered by shotgun sequencing maternal and infant samples [10] (Olm et al., 2021), and KofamKOALA will be used to obtain functional annotations [44] (Aramaki et al., 2020). Similarly, we will assess differences in the microbiome compositions of milk samples. Within the framework of the International Milk Composition Consortium, the milk microbiome will be analyzed via 16S rRNA sequencing. An additional comparison will be performed to determine the associations of the microbiome with maternal health, adverse birth outcomes and infant growth.

#### 2.4.3. Proteomics

The field of proteomics is a high-throughput approach employed to identify the full spectrum of proteins in an organism, tissue, cells or bodily fluid. This approach investigates the functional states of proteins, including protein–protein interactions and posttranslational modifications [61] (Adeola et al., 2017). Navarro et al. (2015) [62] noted that several pathways differed between interventions with glucosamine and chondroitin supplementation and a placebo (Navarro et al., 2015) [62]. 

In the present study, the untargeted quantification of proteins and peptides in prenatal maternal plasma and breast milk samples will be conducted according to a standardized liquid chromatography–tandem mass spectrometry (LC-MS/MS) workflow [45] (Mc Ardle et al., 2022).

#### 2.4.4. Breast Milk Characterization 

Numerous components from breast milk influence the infant microbiota by enhancing the growth of specific bacteria or limiting the growth of others [63] (Boudry et al., 2021). Human milk oligosaccharides (HMOs) interact with gut microbiota by supporting the growth of beneficial bacterial and providing anti-pathogenic effects [64] (Zhang et al., 2021). Another example is lactoferrin, a non-heme iron-binding protein that plays an important role in iron absorption and protection against bacteria [65] (Demmelmair et al., 2017). The impact of maternal supplementation on the interplay of macronutrients, micronutrients and bioactive compounds in human milk is not yet fully understood, highlighting the need for further investigation.

Breast milk samples were distributed to and will be analyzed at multiple laboratories for macronutrients, micronutrients, oligosaccharides, growth factors, immunoglobulins, cytokines, metabolites and microbes. The analyses to be performed on the breast milk and the laboratories employed are summarized in Table 4. 

### 2.5. Human Biomonitoring

Dietary and environmental contaminants, specifically those common in rural, low-income settings (e.g., smoke pollution from cooking, mycotoxins, herbicides, insecticides and pesticides) play important mediating roles and must be considered. The totality of these exposures is the “exposome” [66,67] (Wild, 2005; Miller and Jones, 2014), the study of which is an emerging field with great potential to advance human health research. A biomonitoring analysis will provide insight to determine if the aforementioned exposures act as effect modifiers in the relationship between the provision of BEP and maternal health, birth outcomes and infant growth, as well as any association between the exposure and the relative TL and mtDNAc. A summary of the analyses and the laboratories used are shown in Table 5.

#### 2.5.1. Telomere Length and Mitochondrial DNA

Telomeres protect DNA coding sequences from degradation and prevent the aberrant fusion of chromosomes [81] (Blackburn, Epel and Lin, 2015). In somatic cells, telomeres shorten after each cell division due to the incomplete replication of DNA molecules and maintenance mechanisms that are unable to prevent telomere attrition [82] (Wang et al., 2021). Previous research suggests that short telomeres are associated with cardiovascular disease and mortality [83,84] (Haycock et al., 2014; Wang et al., 2018). In Ghana, prenatal supplementation had no impact on TLs at 4–6 years of age when compared to IFA. However, in Greece, adults receiving a daily combination of vitamin supplements for 6 to 12 months had longer TLs compared to the control group [85] (Tsoukalas et al., 2019).

Mitochondria play a critical role in the production of energy through aerobic respiration, resulting in the formation of adenosine triphosphate [86] (Roger, Muñoz-Gómez and Kamikawa, 2017). The mtDNA is theoretically more susceptible to damage to oxidative stress as it is situated close to the sites of oxidative phosphorylation (e.g., reactive oxygen species). Additionally, mtDNA lacks protection from the histones present in nuclear DNA [87,88] (Tait and Green, 2013; Copeland and Longley, 2014). Mitochondrial dysfunction during the neonatal period and infancy has been related to heart arrhythmia and poor weight gain [89,90] (Gibson et al., 2008; Kohda et al., 2016), whereas in adulthood, mitochondrial dysfunction has been implicated in Alzheimer’s disease and cancer [91] (Druzhyna, Wilson and LeDoux, 2008). In Indonesia, one trial assessed the impact of prenatal MMN supplementation on the mtDNA in venous blood of pregnant women and reported lower post-supplementation mtDNA compared to IFA, indicating improved mitochondrial efficiency [92] (Priliani et al., 2019). 

In the scope of this study, the efficacy of a prenatal BEP supplement and an IFA tablet on newborn relative TL and mtDNAc were compared to the efficacy of IFA alone, and differences in relative TL and mtDNAc across adverse birth outcomes (i.e., SGA, LBW or preterm births) were measured via a real-time PCR method, using whole arterial blood collected from the umbilical cord [68,69,70,71,72] (Cawthon, 2002, 2009; Janssen et al., 2012; Martens et al., 2016, 2020) The results of the effect of the fortified BEP supplementation on the newborn genome were reported in a separate manuscript [14] (Hanley-Cook et al., no date).

#### 2.5.2. Air and Smoke Pollution

In Ouagadougou, the capital of Burkina Faso, 60% of households use biomass-based fuels as their primary cooking fuel [93] (Sana et al., 2019). In 2002, it was estimated that 21,500 deaths in Burkina Faso were attributed to domestic air pollution [94] (WHO, 2007). Exposure to particulate matter air pollution, such as BC, during early life has been linked to adverse pregnancy outcomes, including a LBW [95] (Pedersen et al., 2013), increased cardiovascular morbidity and mortality [96,97] (Brook et al., 2010; Nawrot et al., 2011). In 469 mother–newborn pairs, in utero exposure to particulate matter during the third trimester of pregnancy was linked to a lower placental iodine load, an element that is important for fetal brain development and growth [98] (Neven et al., 2021).

In the present study, using confocal microscopy, the level of BC will be assessed in in whole arterial blood collected from the umbilical cord [73,74] (Saenen et al., 2017; Bové et al., 2019).

#### 2.5.3. Mycotoxins

Mycotoxins are secondary fungal metabolites found on food and feed that can disturb gut microbial homeostasis, metabolism and the integrity of the intestinal barrier [1,3,6,8,10,13,15,17,19,20,26,29,39,40,43,45,46,49,50,51,52,53,55,62,66,70,71,72,74,78,80,86,88,90,92,93,95,97,98,99,100,101,102,103,104,105,106,107,108,109,110,111,112,113] (Hussein and Brasel, 2001; Vidal et al., 2018). In LMICs, mycotoxins pose health risks due to their high abundance and acute intrinsic toxicity [114] (Yacine Ware et al., 2017). Children are vulnerable due to their lower body mass, higher metabolic rate and developing detoxification system [107] (Peraica, Richter and Rašić, 2014). A study in Ethiopia reported a high occurrence of long-tern maternal aflatoxin exposure and an associated risk of poorer fetal growth trajectories [111] (Tesfamariam et al., 2022). In Burkina Faso, limited biological and toxicological food contamination data are available [101] (Kpoda et al., 2022), and regulations or legislation concerning mycotoxins are often not implemented [115,116] (FAO, 2003; Warth et al., 2012).

Maternal and infant whole blood, extracted via VAMS, will be analyzed for mycotoxins using an adapted LC-MS/MS methodology [23] (Vidal et al., 2021).

#### 2.5.4. Environmental Contaminants

Burkina Faso’s economy relies heavily on the agricultural sector, which provides employment for most of the population and generates nearly half of the gross domestic product, yet disease and animal pests cause significant damage to crops. To address this, plant protection products are used to eradicate pests [106] (Ouedraogo et al., 2011), leading to high levels of exposure among the Burkinabè population [117,118,119] (Lehmann et al., 2017, 2018; Son et al., 2018). A study by [101] Kpoda et al. (2022) reported that more than 58% of the food samples collected from Burkinabè markets (i.e., cereals, oilseeds, vegetables and dried fish) contained at least one pesticide. Researchers postulate that exposure to these pesticides increases the chances of miscarriage and other adverse birth outcomes [101] (Kpoda et al., 2022). Previous studies in India and Egypt reported that interventional strategies reduced pesticide-induced oxidative effects [104,109] (Saad-Hussein et al., 2020; Medithi et al., 2022). 

Analyses of environmental contaminants (i.e., herbicides, insecticides and pesticides) in maternal urine will be conducted using LC-MS/MS and LC–high resolution mass spectrometry (LC-HRMS) [75,76] (Gys et al., 2020; Caballero-Casero et al., 2021).

#### 2.5.5. Gut Enteropathogens

The TaqMan array card (TAC) is a 384-well platform that uses primers and probes specific to the targets pre-allocated on the card, allowing for the simultaneous detection of up to 48 targets in one specimen [102,120] (Diaz et al., 2013; Liu et al., 2013). Additional advantages of the TAC assay include its ease of use and excellent reproducibility [121,122] (Weinberg et al., 2013; Diaz et al., 2019). TAC is an ideal platform for multi-pathogen detection in low-resource settings [103,105,121] (Diaz et al., 2019; Moore et al., 2019; Marks et al., 2021). In Nepal, Tanzania and Bangladesh, a TAC analysis detected multiple pathogens with high sensitivity and enhanced the understanding of mixed infections detected in one matrix [78] (Liu et al., 2014).

In this study, a TAC analysis will be performed on maternal and infant fecal samples to assess the pathogen burden, as previously described [77,78] (Liu et al., 2014; Deboer et al., 2018).

#### 2.5.6. Fecal Inflammatory Markers 

##### Calprotectin

Calprotectin is a calcium-binding protein belonging to the S100 series [123,124] (Yui, Nakatani and Mikami, 2003; Jukic et al., 2021) that accounts for 30–60% of the protein content of neutrophils [125] (Dale et al., 1985). Calprotectin interferes in physiological behaviors such as cell differentiation, immune regulation and inflammation [123] (Jukic et al., 2021). The release of calprotectin in the gastrointestinal tract lumen and its excretion in feces are consequences of an inflammatory process that prompts the migration of neutrophils into the gastrointestinal tissue. Calprotectin is therefore a robust and noninvasive marker for intestinal inflammation, whether acute or chronic [126] (Summerton et al., 2002). 

Am enzyme-linked immunosorbent assay (ELISA) will be applied to maternal and infant fecal samples to detect calprotectin since it has demonstrated acceptable intra- and inter batch precision, good recovery and dilution linearity across different concentrations [114] (Whitehead et al., 2013).

##### Short-Chain Fatty Acids

Short-chain fatty acids (SCFAs) are formed in the colon via the fermentation of proteins and non-digestible carbohydrates [127] (Cummings and Macfarlane, 1997). SCFAs are sources of energy for the epithelium and regulate differentiation and proliferation [128,129] (Louis and Flint, 2009; Blad, Tang and Offermanns, 2012). Butyric acid, an essential SCFA, protects against inflammatory bowel disease [130] (Berries et al., 2018), colorectal cancer [53,131] (Scheppach, Bartram and Richter, 1995; Gomes et al., 2018) and cardiovascular disease [86] (Richards et al., 2016). Butyrate regulates the epithelial barrier by organizing proteins, thus reducing the permeability of bacteria [132] (Cani et al., 2008). A study examining dietary effects on gut microbiota found that Burkinabè infants had increased bacterial diversity and higher levels of SCFAs in their feces compared to European children [133] (De Filippo et al., 2010). 

A quantitative analysis of SCFAs will be conducted in maternal and infant fecal samples by means of capillary gas chromatography (GC) coupled with a flame ionization detector (FID) [80] (Toe et al., 2020). 

## 3. Metadata

For all participants, questionnaires and lifestyle and clinical data were collected. In addition, the participants’ dietary intakes were assessed using weekly food-group-diversity questionnaires. Complete details on the metadata collection were described previously [113] (Vanslambrouck et al., 2021).

## 4. Data Quality Control

The MISAME-III field data were collected using SurveySolutions v.21.5 on tablets and synchronized to a cloud-based server weekly. Furthermore, generic validation codes were programmed to limit the entry of implausible values and to improve data quality. Bi-weekly data quality checks were conducted, and missing or inconsistent data were sent back to the field for revision. To ensure the quality of the ultrasound images and the estimations of gestational age (GA), an external gynecologist regularly evaluated 10% of the examinations using a quality checklist and scoring sheet. The trained project workers collected daily data on BEP and IFA compliance in both prenatal study arms via smartphone-assisted personal interviewing programmed in CSPro v.7.3.1. Six supervisors performed monthly lot quality assurance sampling (LQAS) schemes of each home visitor’s work on a random day [8] (Valadez et al., 1996).

## 5. Ethical Considerations

The protocol of this study was approved by the ethics committee of Ghent University Hospital in Belgium (B670201734334) and the ethics committee of the Institut de Recherche en Sciences de la Santé in Burkina Faso (50-2020/CEIRES). An independent Data and Safety Monitoring Board (DSMB), comprising an endocrinologist, two pediatricians, a gynecologist and an ethicist of both Belgian and Burkinabè nationalities, was established prior to the start of the efficacy trial. The DSMB managed remote safety reviews for adverse and serious events at 9 and 20 months after the initiation of enrolment. The MISAME-III trial was registered on ClinicalTrials.gov (identifier: NCT03533712). 

## 6. Strengths and Limitations

The BioSpé sub-study of the MISAME-III project is unique in that compliance to BEP and IFA supplementation was verified by a community-based network of home visitors, leading to high levels of observed adherence. Moreover, quantitative 24 h dietary recalls were conducted to assess whether the daily energy and micronutrient requirements were met by integrating the BEP supplement with the participants’ regular diets, as well as to eliminate the possibility of any dietary substitution [100] (de Kok, Moore, et al., 2021). Furthermore, by collecting diverse biological specimens from mother–infant dyads at various timepoints, we were able to obtain comprehensive -omics data and conduct biomonitoring on the contaminants present in these samples. In conjunction with lifestyle data, this will assist in evaluating the physiological effect of maternal perinatal BEP supplementation. Additionally, it will facilitate the understanding of the intermediary role of environmental contaminants in the relationship between BEP supplementation, maternal health, birth outcomes and infant growth in Burkina Faso.

An additional strength of this study lies in the comprehensive characterization of multi-omics profiles, including exogenous and endogenous exposures. Additionally, the prospective, longitudinal collection of biospecimens minimizes the possibility of measurement errors in the analysis. Our research group’s diverse expertise across numerous disciplines enables a holistic approach to assessing exposures and biological responses. Moreover, to avoid interlaboratory variability and ensure consistency, all laboratory experimentation for each -omics and exposure analysis will be conducted in the same laboratory.

A notable limitation of this study is its sample size, which is insufficient to thoroughly investigate rare diseases or extreme values for continuous traits unless combined with data from other cohorts. Also, the study is monocentric, and the study population is largely homogenous (i.e., rural and African); therefore, its generalizability to the larger population, and other geographical regions and urban settings, is limited.

In conclusion, the BioSpé study will help generate evidence-based health prevention and intervention strategies that improve maternal health, enhance birth outcomes, promote healthy infant growth and address related health issues such as metabolic disorders and accelerated biological aging. In doing so, the results will help us understand mechanistic pathways that will provide valuable insights to inform policy decisions in public health. 

## Figures and Tables

**Figure 1 nutrients-15-04056-f001:**
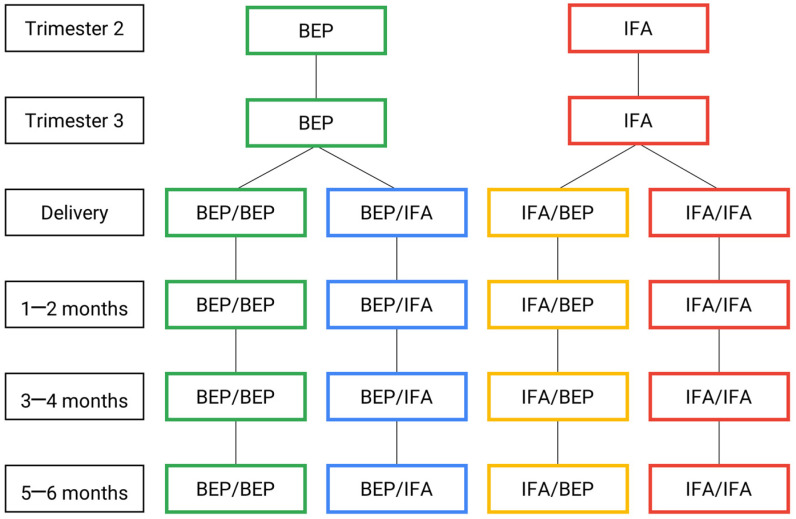
MISAME-III efficacy trial timeline: intervention and control arms. BEP, micronutrient-fortified balanced energy–protein; IFA, iron–folic acid.

**Figure 2 nutrients-15-04056-f002:**
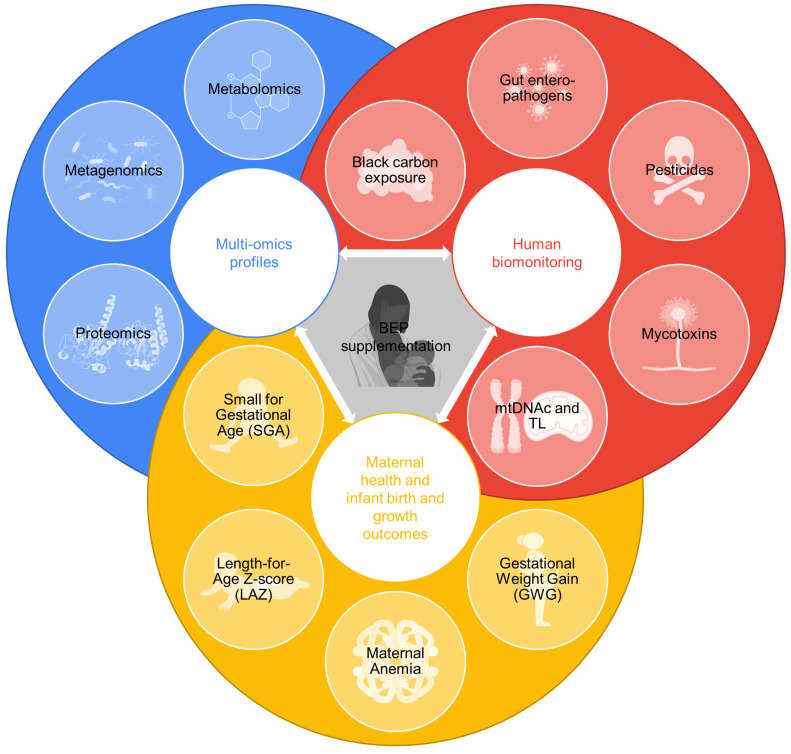
Data produced in the biospecimen sub-study (BioSpé) of MISAME-III. These data will be leveraged to assess the physiological effects of micronutrient-fortified balanced energy–protein (BEP) supplementation on maternal health, infant birth and infant growth outcomes through an analysis of multi-omics profiles, the human biomonitoring of contaminants and measurements of relative telomere length (TL) and mitochondrial DNA content (mtDNAc).

**Table 1 nutrients-15-04056-t001:** List of biospecimens collected for the BioSpé study.

	tri2	tri3	Birth	pn0.5–0.75	pn12	pn34	pn56
1. Plasma (metabolome and proteome)	M	M			M		M
2. Whole blood (exposome and metabolome)	M	M	I		I	I	D
3. Cord blood (relative TL, mtDNAc and black carbon)			I				
4. Breast milk (metabolome, metagenomics and proteome)				M	M	M	
5. Urine (exposome)	M	M					
6. Feces (calprotectin, SCFAs and TAC analysis)	M	M			D	I	D

D—dyad; I—infant; M—mother; mtDNAc—mitochondrial DNA content; pn0.5–0.75—14–21 days after delivery; pn12—postnatal, 1–2 months; pn34—postnatal, 3–4 months; pn56—postnatal, 5–6 months; SCFAs—short-chain fatty acids; TAC—TaqMan array card; TL—telomere length; tri2—trimester 2; tri3—trimester 3.

**Table 2 nutrients-15-04056-t002:** Nutritional values of the BEP supplement for pregnant and lactating women ^1^.

	Mean for 72 g (Serving Size)
Total energy (kcal)	393
Lipids (g)	26
Linoleic acid (g)	3.9
α-Linoleic acid (g)	1.3
Proteins (g)	14.5
Carbohydrates (g)	23.3
Calcium (mg)	500
Copper (mg)	1.3
Phosphorus (mg)	418
Iodine (µg)	250
Iron (mg)	22
Selenium (µg)	65
Manganese (mg)	2.1
Magnesium (mg)	73
Potassium (mg)	562
Zinc (mg)	15
Vitamin A (µg RE) ^2^	770
Thiamin (mg)	1.4
Riboflavin (mg)	1.4
Niacin (mg)	15
Vitamin B5 (mg)	7
Vitamin B6 (mg)	1.9
Folic acid (µg)	400
Vitamin B12 (µg)	2.6
Vitamin C (mg)	100
Vitamin D (µg cholecalciferol) ^3^	15
Vitamin E (mg α-tocopherol) ^4^	18
Vitamin K (µg)	72

^1^ Ingredients: vegetable oils (rapeseed, palm and soy in varying proportions), defatted soy flour, skimmed milk powder, peanuts, sugar, maltodextrin, soy protein isolate, vitamin and mineral complex, stabilizer (fully hydrogenated vegetable fat; mono- and diglycerides). BEP—balanced energy−protein; IU—international unit; RE—retinol equivalent. ^2^ Vitamin A RE, 1 µg = 3.333 IU vitamin A. ^3^ Cholecalciferol, 1 μg = 40 IU vitamin D. ^4^ α-Tocopherol, 1 mg = 2.22 IU vitamin E.

**Table 3 nutrients-15-04056-t003:** Multi-omics profiles investigated in the BioSpé Study (excluding breast milk analyses—see Table 4).

Analysis	Sample Matrix	Volume Collected	Analysis	Analytical Technique	Laboratory	Methodology
Metabolome	Capillary whole blood	10 μL via VAMS	Untargeted	rLC-MS	Sapient Bioanalytics, California, United States	[42] (Villar et al., 2022)
Microbiome profile	Feces	1.8 mL	Untargeted	Quantitative shotgun metagenomics	Stanford University, Stanford, California, United States	[43] (Olm et al., 2021)
				KofamKOALA		[44] (Aramaki et al., 2020)
Proteomics	Plasma	100 μL	Untargeted	LC-MS/MS	Cedars-Sinai Medical Center, California, United States	[45] (Mc Ardle et al., 2022)

LC-MS/MS, liquid chromatography–tandem mass spectrometry; rLC-MS, rapid liquid chromatography–mass spectrometry; VAMS—volumetric absorptive microsampling.

**Table 4 nutrients-15-04056-t004:** Summary of the analyses performed on breast milk.

Analysis	Sample Matrix	Volume Collected	Analysis	Analytical Technique	Laboratory	Methodology
Macronutrients and fat-soluble vitamins	Breast milk	7.2 mL	Untargeted	NIR analysis	University of California Davis, United States	[46] (Smilowitz et al., 2014)
				LC-MS/MS		[47] (Hampel, Dror and Allen, 2018)
Water-soluble vitamins				LC-MS/MS		[47] (Hampel, Dror and Allen, 2018)
				UPLC-MS/MS		[48] (Hampel, York and Allen, 2012)
				Automated immunoassay		[47] (Hampel, Dror and Allen, 2018)
Minerals				ICP-MS		[47] (Hampel, Dror and Allen, 2018)
HMOs				LC-MS/MS	Bode Lab, University of California San Diego, United States	[49] (Kellman et al., 2022)
Proteins				ECL		[50] (Ju, Lai and Yan, 2017)
Metabolomics				rLC-MS	Sapient Bioanalytics, United States	[42] (Villar et al., 2022)
Proteomics				LC-MS/MS	Precision Biomarker Laboratories, United States	[45] (Mc Ardle et al., 2022)
Microbiome				16S rRNA	Baylor College of Medicine, Alkek Center for Metagenomics and Microbiome Research, United States	[51] (Ramani et al., 2018)
Metabolomics (small molecules)			Targeted	LC-MS/MS	Biocrates, Innsbruck, Austria	[52] (Langsdorf et al., 2023)
Metabolomics (lipids and hexoses)			Targeted	FIA-MS/MS	Biocrates, Innsbruck, Austria	[52] (Langsdorf et al., 2023)

ECL, electrochemiluminescence; HMO, human milk oligosaccharide; HPLC, high-performance liquid chromatography; ICP-MS, inductively coupled plasma mass spectrometry; LC-MS, liquid chromatography–mass spectrometry; NIR, near-infrared; QTRAP, quadrupole ion trap; rLC-MS, rapid liquid chromatography–mass spectrometry; rRNA, ribosomal RNA; UHPLC, ultra-high-performance liquid chromatography.

**Table 5 nutrients-15-04056-t005:** Environmental contaminants investigated in the BioSpé study.

Analysis	Sample Matrix	Volume Collected	Analysis	Analytical Technique	Laboratory	Methodology
Telomere length and mitochondrial DNA content	Whole arterial blood from umbilical cord	200 μL	Targeted	Real-time PCR method	Centre for Environmental Sciences, Hasselt University, Belgium	[68,69,70,71,72] (Cawthon, 2002, 2009; Janssen et al., 2012; Martens et al., 2016, 2020)
Black carbon particles	Whole arterial blood from umbilical cord	250 μL	Targeted	Confocal microscopy	Centre for Environmental Sciences, Hasselt University, Belgium	[73,74] (Saenen et al., 2017; Bové et al., 2019)
Mycotoxins	Capillary whole blood	20 μL VAMS	Targeted	LC-MS/MS	Centre of Excellence in Mycotoxicology and Public Health, Faculty of Pharmaceutical Sciences, Ghent University, Belgium	[23] (Vidal et al., 2021)
Pesticides	Urine	4 mL	Targeted	LC-MS/MS	Toxicological Centre, University of Antwerp, Belgium	[75,76] (Gys et al., 2020; Caballero-Casero et al., 2021)
New/emerging contaminants			Untargeted	LC-HRMS		
Multiple Infection Targets	Feces	1.8 mL	Targeted	TAC analysis	Institut de Recherche en Sciences de la Santé, Bobo-Dioulasso, Burkina Faso	[77,78] (Liu et al., 2014; Deboer et al., 2018)
Calprotectin	Feces	1.8 mL	Targeted	ELISA	Institut de Recherche en Sciences de la Santé, Bobo-Dioulasso, Burkina Faso	[79] (Whitehead et al., 2013)
Short-chain fatty acids	Feces	1.8 mL	Targeted	GC-FID	Faculty of Bioscience Engineering, Ghent University, Ghent, Belgium	[80] (Toe et al., 2020)

ELISA, enzyme-linked immunosorbent assay; GC-FID, gas chromatography coupled with a flame ionization detector; LC-MS/MS, liquid chromatography–tandem mass spectrometry; LC-HRMS: liquid chromatography–high-resolution mass spectrometry; PCR, polymerase chain reaction; TAC, TaqMan array card; VAMS, volumetric absorptive microsampling.

## Data Availability

Given the personal nature of the data, data will be made available through a data-sharing agreement. Please contact carl.lachat@ugent.be for any queries. Supporting study documents, including the study protocol and questionnaires, are publicly available on the study’s website: https://misame3.ugent.be (accessed on 18 September 2023).

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
