# Peer review of "A Multi-Omics and Human Biomonitoring Approach to Assessing the Effectiveness of Fortified Balanced Energy–Protein Supplementation on Maternal and Newborn Health in Burkina Faso: A Study Protocol"

_nutrients, 2023, doi:10.3390/nu15184056_

Round 1

Reviewer 1 Report

This manuscript is a study protocol describing the procedures and analyses to be done for a sub study nested within a bigger trial. Overall, the manuscript is written nicely with a lot of needed and useful information.

One note that should be mentioned: Based on the description of the journal focuses on original research, implying research that has been conducted, reviews, and comments. Hence, a study protocol, while not explicitly excluded, is not a typical manuscript for this journal based on their statement. Since this manuscript was forwarded for peer review, I assume the study protocols are not excluded in their portfolio of publications, but it wouldn’t probably be the first journal coming to mind for publishing a study protocol.

Nevertheless, a few things should be addressed before the manuscript can be considered for publication.

Authors:

1.     The “*” is indicating the corresponding authors of the manuscript as per description. Yet, the first 2 authors are indicated by a “*”, while the footnote only includes the contact information for the first and last author. Please revise.

2.     Table 2 shows the concentrations of supplements in the BEP. Based on this, the BEP itself contains iron and folic acid in the same amounts as the IFA placebo. Thus, are the participants in the BEP+IFA group receiving double the amount of these micronutrients than the ones the IFA group? A bit more clarification would be helpful.

3.     P7 L208: What is the VAMS device? If this is a commercially available device, please add same, model, manufacturer info. This should always be added to all of the equipment mentioned.

4.     P9, 2.4.1.: Given the targeted metabolomics you are describing, you may want to Semba et al. 2016, as they used also Biocrates target metabolomics to show significant differences in amino acid and some lipid-metabolite concentrations in stunted vs. non-stunted children. Further, Hampel et al. 2019 found similar trends in human milk collected from apparently healthy mothers vs mother with stunted infants.

5.     P9 L334 states that “All metabolomics analyses will be conducted by rLC-MS using a previously developed method.” Based on the information in your tables this is incorrect. I deduct given the information in the text that all untargeted metabolomics are done using the same method regardless of the matrix (e.g., plasma/serum vs milk). But targeted metabolomics is done differently by using the Biocrates platforms. Please review and edit accordingly.

6.     Table 3: This is a nice overview, but if you could add the references to each method, the reader can easily look for the methods used. After all this is a study protocol, so the methods are the heart and soul of this manuscript. The more details are packed in here, the less have to be added in the outcome papers.

7.     Table 4: While I understand that Table 3 shows the analyses done on blood and stool, and Table 4 is reserved for human milk analyses, why did you opt to change the format for the tables as they are basically providing the same information? Especially since Table 5 is presented in the same format as Table 3. All Tables should have the references included for each method and should be presented in the same format.

8.     Table 4: Only the Biocrates row includes more details about the compound classes that are analyzed with their kit. Since you are not going into the same details anywhere else, I would remove those and recommend adding the name of the assay. You can add a link or a reference for it as well, just like all the others approaches. Otherwise, you should mention the same for all other analyses as well, which would make for a very non-readable table.

9.     P15, Metadata: This section is extremely short. I assume the noted reference includes all the details for this metadata collection? Please be a bit more specific. Compared to everything else this appears as an afterthought. Just a little sentence such as “Details on metadata collection have been described previously (refXXX).” clarifies to the less familiar reader where to find all the info if needed.   

Reviewer 2 Report

I was confused by how complicated the protocol seemed - it looked like a simple clinical trial protocol but there there were so  many sub variants and factors - I got lost. 

What exactly are you trying to measure finally - i thought it was weight gain and Hemoglobin - but is looks so much more complicated than that

Reviewer 3 Report

Thank you for submitting this manuscript to Nutrients. The manuscript is well written and described a sub study of MISAME-III trial. I appreciate the completeness of the coverage of listed biological data collection and proposed analysis. However, clear differentiation between the proposed sub study and the MISAME-III trial would be useful.

Round 2

Reviewer 2 Report

thank you for the revisions - it makes a lot of sense

It will be useful to a specific target group so that is why I am agreeing to accept this paper